# Integration of Nanomaterials and Bioluminescence Resonance Energy Transfer Techniques for Sensing Biomolecules

**DOI:** 10.3390/bios9010042

**Published:** 2019-03-16

**Authors:** Eugene Hwang, Jisu Song, Jin Zhang

**Affiliations:** 1School of Biomedical Engineering, University of Western Ontario, 1151 Richmond St., London, ON N6A 5B9, Canada; ehwang8@uwo.ca (E.H.); jsong254@uwo.ca (J.S.); 2Department of Chemical and Biochemical Engineering, University of Western Ontario, 1151 Richmond St., London, ON N6A 5B9, Canada

**Keywords:** bioluminescence resonance energy transfer, fluorescent nanomaterials, fluorescent nanobiosensors, quantum dots

## Abstract

Bioluminescence resonance energy transfer (BRET) techniques offer a high degree of sensitivity, reliability and ease of use for their application to sensing biomolecules. BRET is a distance dependent, non-radiative energy transfer, which uses a bioluminescent protein to excite an acceptor through the resonance energy transfer. A BRET sensor can quickly detect the change of a target biomolecule quantitatively without an external electromagnetic field, e.g., UV light, which normally can damage tissue. Having been developed quite recently, this technique has evolved rapidly. Here, different bioluminescent proteins have been reviewed. In addition to a multitude of bioluminescent proteins, this manuscript focuses on the recent development of BRET sensors by utilizing quantum dots. The special size-dependent properties of quantum dots have made the BRET sensing technique attractive for the real-time monitoring of the changes of target molecules and bioimaging in vivo. This review offers a look into the basis of the technique, donor/acceptor pairs, experimental applications and prospects.

## 1. Introduction

Bioluminescence resonance energy transfer (BRET) sensors offer a high degree of sensitivity and reliability in a procedure that is both easy to perform and relatively expensive. BRET sensors utilize bioluminescence generated from a luciferase enzyme found in the sea pansy in order to donate energy to a fluorescence molecule that will emit a measurable quantity of fluorescence [1,2,3]. This system can be used to monitor an easily detectable light signal in real time with a high degree of sensitivity. BRET sensors present a less invasive and easier way to perform the modification of fluorescence resonance energy transfer (FRET), which requires an external light source to initiate the fluorescence transfer [4]. In studies that have previously utilized FRET, BRET can be an effective alternative. For example, in a study by Li et al., a FRET dye-labeled probe was used to improve single-base mismatch discrimination in DNA detection [5]. This technique is limited to in vitro applications due to the requirement of external illumination. By utilizing a BRET probe in the place of FRET, this DNA detection technique can be used in live animal models, thereby expanding the potential for this technique. FRET has been widely used in detecting protein-protein interactions [5,6,7,8,9,10]. One such study encountered problems in analyzing protein proximity in the endoplasmic reticulum of cells due to the overlap of the FRET emission wavelength and highly variable cellular autofluorescence [5]. This problem, termed bleed-through, was mitigated by the engineering of a new FRET pair which significantly increased the signal to noise ratio. However, this new FRET pair compromised its signal intensity. Such a problem could potentially be solved using BRET instead, which has been proven to mitigate the effects of cellular autofluorescence [11,12].

No longer requiring an external light source has made BRET attractive for use in biological systems, which may be susceptible to tissue autofluorescence and photobleaching. Background emission from tissue, otherwise known as autofluorescence, was found to be a limiting factor in the sensitivity of reporters [13,14]. Chemiluminescent resonance energy transfer (CRET) is another non-radiative energy transferring process in which the oxidization of a chemiluminescent compound excites an acceptor fluorophore [15,16,17,18,19]. The major advantages for those systems (CRET and BRET) are low background noise and high excitation efficiency, as there is no need for an extra excitation source, typically a high-energy laser. In this way, in particular, a quantum dots (QDs) based BRET system has showed great potential in deep tissue imaging with a low harmful effect on the body. Thus, the use of bioluminescent reporters has become increasingly attractive in biological applications.

BRET is a natural phenomenon involving the non-radiative energy transfer between a bioluminescent donor molecule and a fluorescent acceptor molecule. When the bioluminescent donor, typically an oxidative luciferase enzyme such as NLuc, emits bioluminescent energy, which excites the fluorescent acceptor and increases its emission [14,20], this is known as resonance energy transfer, and only occurs when the two proteins are within 10 nm (Figure 1). The changes in ratio of the acceptor to donor emissions are monitored, which are useful in studying protein-protein interactions (PPI) since the mechanism depends on distance [4,14].

FRET and, by extension, BRET, is a distance dependent mechanism. The efficiency of the energy transfer (*E*) is defined as the quantum yield of the energy transfer transition [21]. The efficiency depends on the distance between the donor and acceptor, usually up to 10 nm. In addition, the emission spectrum of the donor should overlap with the acceptor’s absorption spectrum. The donor emission dipole moment and the acceptor absorption dipole moment should be relatively oriented [22,23]. The efficiency is calculated as follows:(1)$$E=\frac{R_{0}^{6}}{R_{0}^{6}+r^{6}}$$
where *r* is the distance between the donor and acceptor and *R*_0_ is the Förster distance of the donor acceptor pair which is the distance where *E* = 50%.

The Förster distance is given by the following equation:(2)$$R_{0}=0.21{\left[\kappa^{2}Q_{D}n^{-4}J\left(\lambda \right)\right]}^{\frac{1}{6}}$$
where *J*(*λ*) is the spectral overlap between donor emission and acceptor absorption, *Q_D_* is the quantum yield of the donor, *n* is the refractive index of the medium, and *κ*^2^ is an orientation factor related to the relative orientation of the donor emission and acceptor absorption dipole moments.

BRET has revolutionized the detection of small molecules and PPIs. Some techniques utilize a quenching mechanism whereby the acceptor molecule accepts the fluorescence of the donor molecule and quenches the signal. In one particular sensor, gold nanoparticles were used to quench the bioluminescence from RLuc [24]. When the sensor was bound to glucose, the gold nanoparticles could no longer quench the signal, therefore bioluminescence was restored (Figure 2). The intensity of the signal could then be correlated to the concentration of the glucose in a sample.

There exists a variety of bioluminescence donors, many of which are derived from naturally occurring enzymes in animals such as the firefly or the sea pansy. Among the different donors, each have clear advantages and disadvantages in certain applications. Therefore, certain donors are considered for certain tasks according to their properties. This review aims to highlight some of the differences between the donors and how they can be applied in BRET constructs.

Typically, BRET sensors utilize fluorescent proteins [3,5,25,26,27] or organic dyes [28,29,30] as acceptors. However, recent studies report the use of nanoparticles such as quantum dots (QD) to be effective. QDs have the advantage of adjustable emission depending on size, superior brightness, high photostability, and multiplexing [31,32]. Fluorescent semiconductor quantum dots were previously limited in their application for in vivo imaging due to requiring excitation from an external source of light [5,6,7,8]. Combining QDs with luminescence provided by bioluminescent proteins has provided opportunities for in vivo imaging [7]. This review will offer some insight into the many applications of BRET-QDs.

## 2. Bioluminescent Proteins

Table 1 provides a summary on the different bioluminescent proteins. The detailed information of each bioluminescent protein has been described as follows.

### 2.1. Aequorin

First discovered in the jellyfish *Aequorea victoria*, aequorin is a 22 kDa photoprotein that emits blue light at 469 nm when exposed to its substrate, coelenterazine [33,34]. Given its high sensitivity for calcium, aequorin is most often used to detect calcium concentration from a single cell by expressing it using recombinant aequorin [35,36,37]. Typically, aequorin is recombined with polyols to increase its stability [37]. Addition of coelenterazine to the medium allows it to passively diffuse into the cell, and aequorin emits blue light in proportion to the calcium levels within the cell [34]. However, aequorin gives a low light quantum yield compared to other bioluminescent proteins [34]. Furthermore, its substrate, coelenterazine, has been shown to be unstable and have poor biodistribution [38].

### 2.2. Bacterial Luciferase

Bacterial luciferase (Lux) consists of two subunit: alpha, which is 40 kDa, and beta, which is 35 kDa; it emits blue light, which peaks at 490 nm [39,40]. Lux is an ATP dependent luciferase and requires oxygen and NADPH as cofactor, in order to work on its substrate: long-chain aliphatic aldehydes and flavin mononucleotide (FMNH2) [39,40]. It is most often used as a bacterial reporter, more specifically in luminous bacteria for an autonomous bioluminescence oxidation reaction [40]. The long-chain aliphatic aldehydes have shown to be able to freely diffuse through the cell membrane and have high binding affinity for Lux [39]. Like aequorin, Lux demonstrates a poor light quantum yield as well as lack of thermostability [41]. Furthermore, studies are limited to the luminous bacteria due to the cytotoxicity of aldehydes [42].

### 2.3. Firefly Luciferase

Firefly luciferase (Fluc), the first luciferase to be discovered, is a 61 kDa protein that emits blue light at 562 nm when exposed to its substrate, D-luciferin [43,44,45]. Similar to Lux, Fluc is ATP-dependent and requires the presence of co-factors, oxygen and magnesium, in order to complete its reaction with D-luciferin [44,45,46]. Since its discovery, Fluc has been used in various fields as a biosensor through recombination with another protein of interest and as ATP sensor, taking advantage of its ATP-dependency [47,48,49]. Fluc demonstrates high light quantum yield making it superior in comparison to Lux and aequorin [45]. Nonetheless, disadvantages involving Fluc as well as D-luciferin has emerged. Although D-luciferin had been presumed to have good biodistribution given its ability to cross the blood-brain barrier and blood-placental barrier, recent studies have discovered its low tissue permeability, which results in heterogenous biodistribution [38,50,51]. Furthermore, D-luciferin has low affinity for Fluc, which may result in false negative signals [38]. Moreover, Fluc and D-luciferin only gives a single imaging signal limiting studies to a single molecular event or a single population of cells [38]. Finally, the large size of Fluc may lead to steric hindrance when used as a recombinant protein [46].

### 2.4. Renilla Luciferase

Renilla luciferase (Rluc), first discovered in sea pansy, *Renialla reniformis*, is a 36 kDa protein that emits blue light at 480 nm when worked with its substrate, coelenterazine [52,53,54]. Rluc is often used as a marker for gene expression in mammalian cells and as a biosensor when recombined with a protein of interest [53,55]. As it originates from non-mammalian cells, the gene sequence for Rluc includes codons that are uncommon in mammalian cells, which limits Rluc expression in the cells [52]. Furthermore, Rluc has shown to have low enzymatic turnover and quantum yield when compared to Fluc [54,56]. In addition, the problem of instability and poor biodistribution of coelenterazine remains [38].

### 2.5. Gaussia Luciferase

Gaussia luciferase (Gluc) is similar to Rluc as they both have emission peak at 480 nm, are ATP-independent, and work on the same substrate, coelenterazine [46,54,56]. On the contrary, Gluc is smaller than Rluc at 19.9 kDa and originates from *Gaussia princeps* eliminating the problem of uncommon codons [56]. Gluc has been used as a bioluminescent label for in vitro hybridization assay as well as a biosensor through recombination with another protein of interest [46,57]. Gluc is naturally secreted by the cells, which allows for it to be detectable in cell medium [56,58]. It has also been characterized to be more sensitive compared to Fluc and Rluc [58]. Despite its sensitivity, quantum yield remains low and problems involving coelenterazine ensues [38,54,59].

### 2.6. Vargula Luciferase

Vargula luciferase (Vluc), also known as *Cypridina* luciferase, is a 62 kDa protein that emits blue light at 460 nm when worked with its substrate, vargulin, also known as *cypridina* luciferin [60,61,62]. Similar to Rluc, Vluc has been used as a marker for gene expression in mammalian cells as well as a biosensor when recombined with another protein of interest [61,63]. Also similar, to Gluc, Vluc is also naturally secreted by cells, and it is detectable in cell medium [63]. One advantage of Vluc over other luciferases is its glow-type bioluminescence compared to flash-type exhibited by other luciferases [62]. Previously Vluc has been shown to be difficult to express and purify from bacterial systems; however, the issue has been successfully addressed by using a truncated derivative of Vluc, which shows a higher degree of expression and purification while retaining its enzymatic activity [62].

### 2.7. Metridia Luciferase

Like Rluc and Gluc, *Metridia* luciferase also emits blue light at 480 nm, is ATP-independent, and works on the same substrate, coelenterazine [54,57,64]. Smaller than Gluc but bigger than Rluc, *metridia* luciferase is 24 kDa protein [64]. Similar to all luciferase, *metridia* luciferase has been used as a biosensor by recombining it with another protein of interest [64]. *Metridia* luciferase is also naturally secreted by cells similar to Vluc and Gluc [57]. In addition, its low molecular mass serves as an advantage in recombination [64]. However, *Metridia* luciferase demonstrates low quantum yield, and the disadvantages of coelenterazine discussed before stays relevant [38,54].

### 2.8. Nano Luciferase

Nano Luciferase (Nluc) is a recently developed luciferase that uses furimazine as a substrate to emit blue light at 460 nm [65,66,67]. It is the one of the smallest luciferase to be characterized at 19 kDa and has the one of the brightest bioluminescence to date [65,67]. Not many studies using Nluc have been published compared to other bioluminescent proteins as the molecule is fairly new; the published studies typically use Nluc as a biosensor through recombination with their protein of interest [67,68]. Nluc exhibits glow type bioluminescence, similar to Vluc, with long half-life of approximately 2 h [65,67]. Furthermore, furimazine has shown to exhibit lower background noise when compared to coelenterazine [46].

## 3. QD-Based Biosensors by Applying Bioluminescent Resonance Energy Transfer Techniques

Photoluminescent QDs are rapidly becoming popular choices for use in biomedical applications such as labeling, bioimaging, and biosensing. QDs are particularly appealing due to their high photostability, continuous absorption spectra, and size-dependent fluorescence [31,32,69,70,71]. QDs are typically synthesized as hydrophobic and therefore require a number of modifications in order to be suitable in biological environments. Modifying QDs to be suitable for solubility in water results in a decreased quantum yield and therefore requires surface modifications [72,73]. There exist three strategies to make QDs water soluble: ligand exchange, silanization, and encapsulation.

In ligand exchange, the original hydrophobic coating is replaced by a water-soluble bifunctional molecule. Once attached to the QD surface, a hydrophilic tail makes the QDs able to bioconjugate, usually with other surface groups such as thiol, amine and carboxyl [74].

Silanization is an extension of ligand exchange where the QD is coated in a silica shell which is non-toxic, chemically inert and optically transparent. The silica shell protects the QD from oxidization and provides a matrix which enhances stability in the environment. The silica is biocompatible and can be functionalized for bioconjugation [75,76,77].

Encapsulation utilizes different carriers such as amphiphilic polymers, polymeric microbeads, and liposomes [78,79,80]. These coating molecules have hydrophobic and hydrophilic units, therefore can interact strongly with the QD surface and the aqueous outside environment [78]. Liposomes are particularly popular due to their porous spherical structure and high loading capacity [25,79]. However, they are limited due to susceptibility to temperature and pH changes [81].

There are two primary approaches to bind biomolecules onto the surface of QDs. Non-covalent which is mediated by interactions between the biomolecules and the QD surface, and covalent linking which is achieved through chemical reactions of molecular surface groups [80,82].

Non-covalent binding is achieved through two types of interaction; electrostatic interaction between oppositely charged molecules and high affinity secondary interactions. QDs are negatively charged on the surface, therefore can be electrostatically coupled to positively charged proteins [80,83]. High affinity secondary interactions are interactions between functional groups on the surface of the QD and the biomolecule. Biotin-avidin is a commonly utilized combination, though limited by the increased size of the product [83,84]. The interaction between His-tagged biomolecules and Ni-NTA is widely used in bioconjugation and often used for developing a QD probe for performing Western blot analyses [83,84].

Covalent binding involves the reaction between functional groups on QDs and biomolecules. Crosslinkers can be used to bind the molecules. Zero-length crosslinkers such as EDC and DCC are used because they will not add anymore atoms [11]. Carboxylic functionalized QDs are among the most popular due to the abundance of free amine groups on proteins which is can conjugate with [12]. The most popular method utilizes EDC which mediates the formation of an amide bond between the carboxyl on QDs and amines on biomolecules [11,85].

### 3.1. QD-Based BRET Sensor for Detecting Biomolecules

Carboxylated quantum dots (Qd-625) were conjugated to a DNA probe (Qd-D-P) while oligonucleotide probes were conjugated to Rluc (Rluc-P). The sensing scheme uses the two antisense oligonucleotide sequences which will anneal adjacent to each other in a head-to-head fashion when the target is present (Figure 3). The maximum BRET signal was obtained by optimizing the spacing between the Rluc-P and Qd-D-P when hybridized to the target. Optimization was done using oligonucleotide targets, which create various separation distances. The optimal separation distance was found to be 15 nucleotides. Hybridization time between labeled probes and target was studied and found the optimal time to be 5 min and 35 min. In conclusion, an on-type, BRET-based sensing platform incorporating Rluc and QD with a 5-min detection time of a nucleic acid target in vitro was developed. The method was also determined to be highly sensitive (detection limit of 0.54 pmol) and selective against mismatch targets [86].

Near-infrared region optical detection of apoptotic cells was achieved using BRET-coupled annexin V-functionalized quantum dots. A recombinant protein with Rluc and annexin V was conjugated to glutathione-coated CdSeTe/CdS QDs (Figure 4A). Annexin V recognizes phosphatidylserine (PS) on the outer monolayer of the membrane of apoptotic cells and binds to it in the presence of Ca^2+^ ions (Figure 4B). The QDs act as a probe for detecting apoptotic cells with a peak emission at 830 nm and a high quantum yield of 61% in aqueous solution. This method was presented as a simple, rapid, and efficient method for synthesizing a BRET-induced NIR emission probe and with its low phototoxicity could prove to be useful in highly sensitive detection of apoptotic cells in vivo and in vitro [87].

A general BRET homogenous immunoassay was developed for the determination of small molecules. This assay is based on QDs and Rluc which produce variable energy transfer in the presence of different concentrations of free fluoroquinolones (FQs). In the absence of free FQs, QDs conjugated to norfloxacin (QD-NOR) are recognized by a single-chain variable fragment (scFv), which is conjugated to Rluc, and are able to produce an energy transfer (Figure 5). Otherwise, the presence of Free FQs prevents the Rluc and QDs from producing and energy transfer. Similar results for cross-reactivity to seven representative FQs were found when compared to an enzyme-linked immunosorbent assay (ELISA). The LOD of the QD-BRET immunoassay was 2.54 ng/L with a linear range which covers 4 orders of magnitude (0.023 ng/mL to 25.60 ng/mL). The use of QDs enables the flexibility of more choices for donor substrates given the wider excitation range of the QDs. The authors noted that by replacing the target of interest, the immunosensor could be used with a variety of other small molecules and could open up the possibility of multiplex detection using different QDs [88].

In one particular study, CdSe/ZnS core-shell quantum dots (QD705) were conjugated to Nluc (Figure 6). This construct was used to image tumors by conjugating it to cyclic arginine-glycine-aspartic acid (cRGD) peptides, which were selected due to their having a strong affinity for integrin α_v_β_3_ which are known to be expressed on various tumor types. When intradermally injected into the hind paw of a mouse, the popliteal lymph nodes could be visualized by bioluminescence at 5 min post injection (p.i.) without any background signal. To demonstrate tumor targeting capabilities, the sensor was injected into mice with integrin expressing U87MG human glioblastoma cell tumors. After 2 h p.i. organs were harvested and ex vivo imaging found a noticeably visible signal in the tumors injected with QD-Nluc-cRGD. When compared to traditional fluorescence techniques, bioluminescence demonstrated higher sensitivity due to the distance dependent relationship of BRET components. Using the BRET conjugate, the tumor to background ratio was exceptionally high (>85) [89].

### 3.2. QD-Based BRET Sensor Used in Bioimaging In Vivo

BRET-QD nanoparticles were applied to in vivo lymphatic imaging in mice. QD655 covalently linked to Luc8 protein, an eight-mutation variant of Rluc, were intracutaneously injected into 10 weak old normal athymic female mice at different sites, the chin, ear and paws. After 5 min, the imaging was carried out (Figure 7). Using BRET-QD655 has the advantage over traditional bioluminescence imaging (BLI) in that NIR light emission is more favorable in tissue penetration. All lymph nodes were visualized when injected with BRET-QD655 constructs and since there is no excitation light, the BRET signal more accurately represents the concentration of the quantum dots in the lymph nodes, leading to better quantitative imaging [90].

The use of QD nanoparticles as a tool for non-invasive investigation of mammalian spermatozoa was explored. 655 nm emitting CdSe/ZnS QD nanoparticles were conjugated to nona-Arginine R9 peptide which facilitates cellular internalization. The QDs were linked to RLuc and used to label boar spermatozoa and were assessed for changes in sperm motility, viability, and fertilizing potential. In vitro assays concluded no adverse effect of the BRET-QD on the spermatozoa. The results suggest the strong potential for a novel imaging technique for tracking BRET-QD labeled spermatozoa to better understand sperm migration within the female genital tract [91].

### 3.3. Other QDs-Based Applications

Previous studies have demonstrated the potential for low level laser for treatment (LLLT) of Alzheimer’s disease. LLLT has proven to be effective in mitigating amyloid-β peptide induced oxidative stress and inflammation. However, this technique is limited by penetration depth in live animals and human subjects. In a study by Bungart et al., quantum dots in a BRET conjugate were employed as a source of near infrared light (NIR) [92]. 800 nm-emitting BRET-QDs activated with colenterazine were used to pre-treat primary cortical rat astrocytes. When the astrocytes were exposed to 5 μM amyloid-β for 2 h, light treated cells had reductions in amyloid-β induced superoxide anion production (Figure 8) and inflammatory marker expression to untreated control levels. Using BRET-QDs or coelenterazine individually resulted in reductions that were not statistically significant. NIR light cannot penetrate further than a few centimeters [92] therefore BRET-QDs offer a promising alternative for non-invasive LLLT for Alzheimer’s disease.

## 4. Biocompatibility

Quantum dots cores consist of various metal complexes which have raised questions about their application in a biological setting [93]. This concern is commonly resolved by adding organic coatings such as methoxy-polyethylene glycol (PEG) to the surface in order to increase biocompatibility [94]. Commonly used quantum dots include CdTe and CdSe and all of these metals are known to be toxic to humans when exposed upon degradation of the quantum dots [95,96,97,98,99]. There is lack of biocompatibility data for quantum dots and little research done on the effect on humans. Part of the reason can be attributed to the various factors related to physicochemical properties of quantum dots like size, charge, concentration, outer coating and mechanical stability [100,101]. Various studies have found adverse effects due to quantum dots [102] and some have found little to no adverse effects, given some modifications [103,104]. Studies have determined that quantum dots have a significantly long half-life and the degradation of fluorescent particles taking almost 12 weeks in the liver [94,105]. This uncertainty in the biocompatibility warrants further investigation, particularly in human subjects.

## 5. Conclusions

Owing to the quick development of luminescence nanomaterials and bioluminescence proteins, BRET sensing technology benefits from the conjugation of these two cutting edge materials. QDs offer unparalleled function in their high intensity, stability, broad emission and narrow emission spectra, size and tunable emission. Along with a wealth of bioluminescent proteins with properties to suit a wide range of applications, BRET-QD sensors demonstrate a high potential for applications ranging from protein-protein interaction to biomolecule detection. Though the biocompatibility of QDs has yet to be proven, there are existing techniques to increase their viability in biological settings.

In conclusion, BRET in conjunction with quantum dots is very promising for in vivo applications despite uncertainty in its biocompatibility. It is a far more effective alternative to traditional FRET methods, and with sufficient research can be used for a wide range of applications in imaging and nanomedicine.

## Figures and Tables

**Figure 1 biosensors-09-00042-f001:**
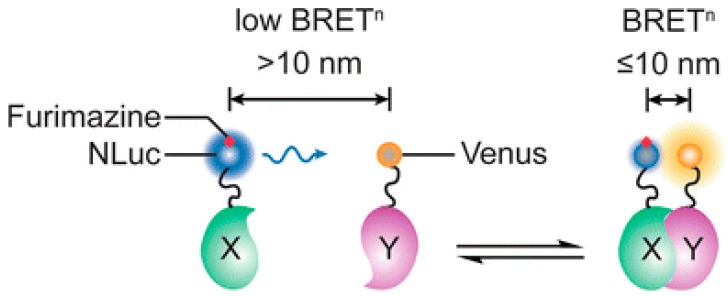
Schematic of a NLuc based BRET (BRET^n^) design. Nluc is the donor fluorphore and Venus is the acceptor fluorophore. Nluc and Venus are fused to their respective proteins of interest (X and Y). BRET signal is detected when the proteins are in close proximity. Reprinted by permission from Springer Nature [14]. Copyright 2016.

**Figure 2 biosensors-09-00042-f002:**
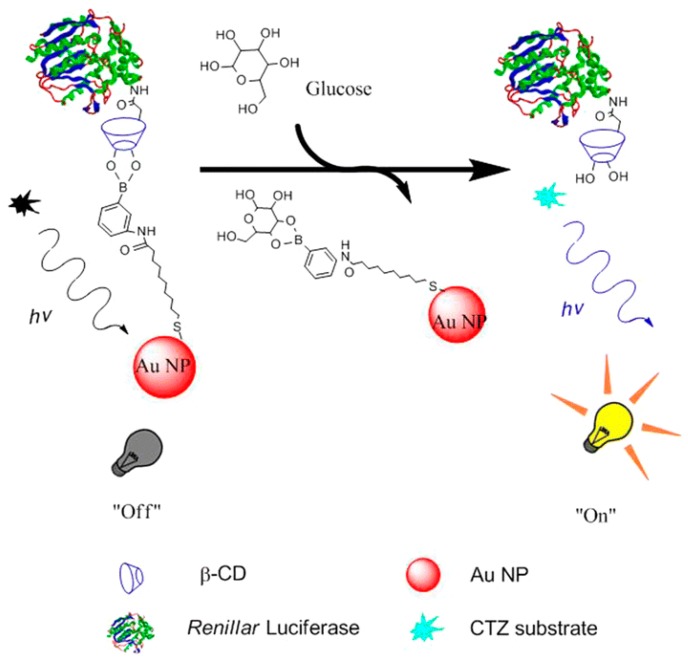
Schematic illustration of bioluminescence quenching-based nanosensor in glucose sensing. Reprinted with permission from [24]. Copyright Springer Nature 2017.

**Figure 3 biosensors-09-00042-f003:**
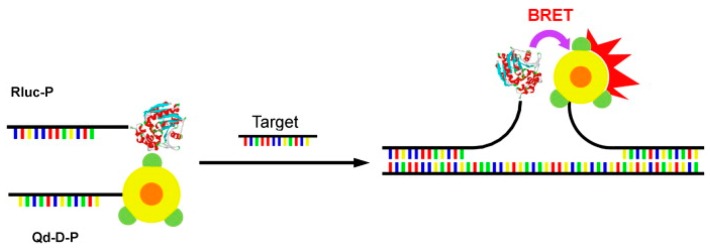
Schematic of BRET based on-type sensing system. The Rluc-P and Qd-D-P hybridized to the target in head-to-head fashion permitting BRET between Rluc and Qd. Reprinted from [55], with permission from Elsevier.

**Figure 4 biosensors-09-00042-f004:**
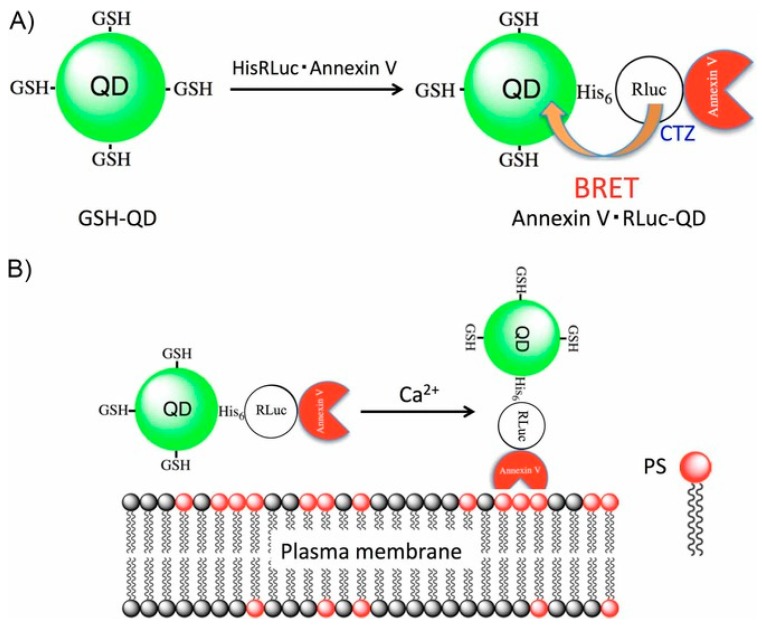
(**A**) Synthetic method for the preparation of recombinant protein (HisRLuc-annexin V)-conjugated QDs (annexin V-Rluc-QDs). (**B**) Schematic representation for the binding of annexin V-RLuc-QDs to PS on plasma membrane of apoptotic cells in the presence of Ca^2+^ ions. Reprinted with permission from [56]. Copyright 2017 John Wiley and Sons.

**Figure 5 biosensors-09-00042-f005:**
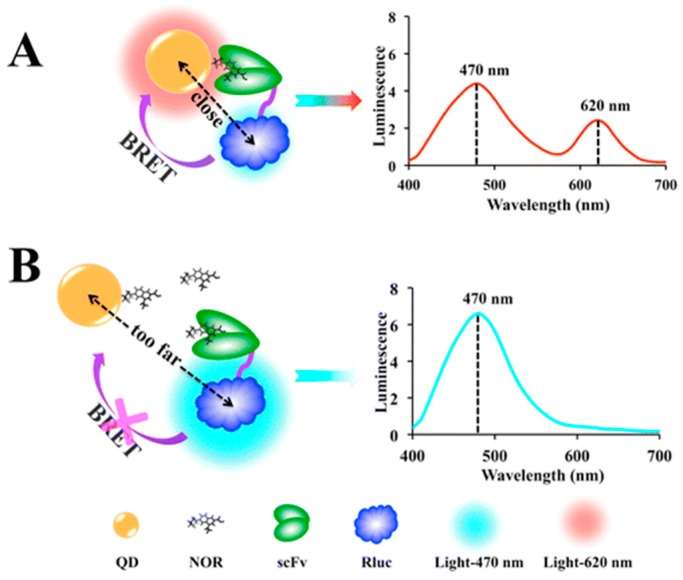
(**A**) In the absence of free FQs, QD-NOR is recognized by scFv-Rluc and the Rluc and QDs are in close proximity; energy is released from the substrate and transferred to the QDs via BRET. (**B**) In the presence of free FQs, the scFv-Rluc binds to the free FQs and the distance between the Rluc and QDs is too far to realize energy transfer. Reprinted with permission from [57]. Copyright 2016 American Chemical Society.

**Figure 6 biosensors-09-00042-f006:**
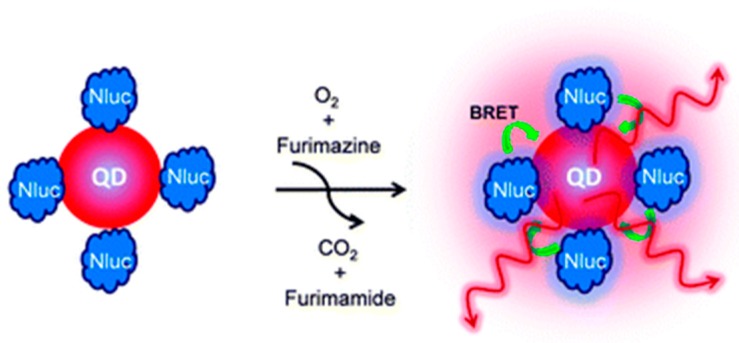
QD-Nluc-cRGD conjugate. Chemical communications by Royal Society of Chemistry (Great Britain) Reproduced with permission of ROYAL SOCIETY OF CHEMISTRY in the format Journal/magazine via Copyright Clearance Center.

**Figure 7 biosensors-09-00042-f007:**
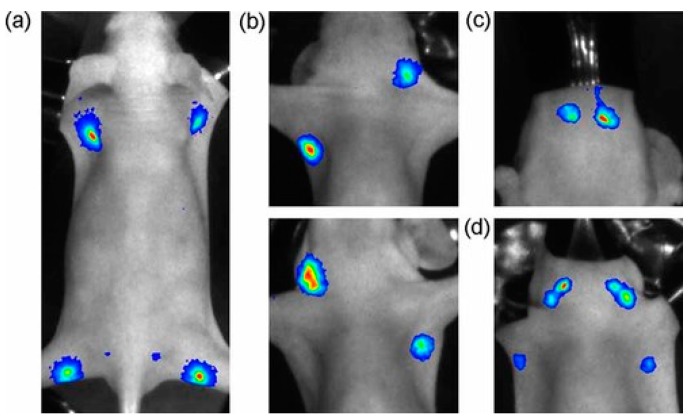
BRET lymphatic images of different lymphatic basins. BRET-QD655 were injected at all four paws (**a**), the ear and forepaw (**b**), the chin (**c**), or five different sites (**d**) both forepaws, both ears and chin). Reprinted with permission from [59]. Copyright 2011 John Wiley and Sons.

**Figure 8 biosensors-09-00042-f008:**
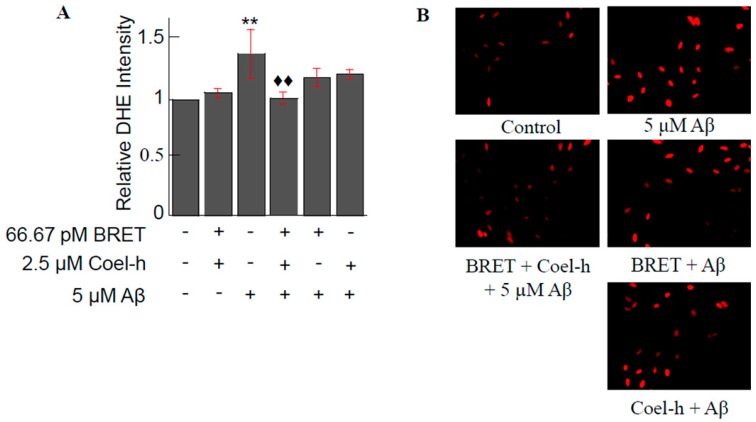
BRET-QD with coel-h pretreatment on Aβ-induced superoxide anions production. (**A**) Aβ treatment increased the fluorescent intensity of DHE by 40% as compared to control. The pretreatment of BRET-QD (66.7 pM) and coel-h (2.5 μM) suppressed the Aβ-induced increase in superoxide anion production back to that of control. Both the BRET-QD and Coel-h alone pretreatment lowered Aβ-induced superoxide anion, but not with a statistical significance. (**B**) Fluorescence images representing each experimental group. Each experiment was repeated three timesas compared to 5 μM Aβ. Data of each treatment group were normalized by the control for each independent experiment prior to statistical analysis. Reprinted from [92], with permission from Elsevier.

**Table 1 biosensors-09-00042-t001:** Summary of Bioluminescent proteins.

Name	Size (kDa)	Emission (nm)	Substrate
Aequorin	22	469	Coelenterazine
Bacterial luciferase (Lux)	Alpha subunit: 40 Beta subunit: 35	490	FMNH_2_ long-chain aliphatic aldehyde
Firefly luciferase (Fluc)	61	562	D-luciferin
*Renilla* luciferase (Rluc)	36	480	Coelenterazine
*Gaussia* luciferase (Gluc)	19.9	480	Coelenterazine
Vargula luciferase (Vluc) or *Cypridina* luciferase	62	460	Vargulin (Cypridina luciferin)
*Metridia* luciferase	24	480	Coelenterazine
Nano luciferase (Nluc)	19	460	Furimazine

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
