# Peer review of "Integration of Nanomaterials and Bioluminescence Resonance Energy Transfer Techniques for Sensing Biomolecules"

_biosensors, 2019, doi:10.3390/bios9010042_

Reviewer 1 Report

The manuscript entitled “Integration of Nanomaterials and Bioluminescence Resonance Energy Transfer Techniques for Sensing Biomolecules” by Ji Su Song and Jin Zhang presented BRET-QD technology and its application in biomaterial sensing and imaging. In detail, authors described some of bioluminescent proteins as a source of light donor. Applications such as detection of DNA mismatch, small molecule and apoptotic cells were illustrated as well as in vivo and in vitro imaging of spermatozoa, tumors, lymphatic. However, this manuscript shows lack of accuracy and the information is not comprehensive to meet the standard for publication to Biosensors as a “Review”. 

Figures 1 and 2 and the corresponding description are very similar with previous review presented by Fei Ma, et al. (Figures 7 and 8; J. Mater. Chem. B, 2018, 6, 6173-6190).

This manuscript poorly referenced in context of bioluminescent proteins, in particular Table 1 such as

1)      The list of luciferases is not comprehensive compared with a list presented by Gokhin S., et. al. in 2017 (http://dx.doi.org/10.1039/9781782626770-00117).

2)      Author failed to include a luciferase, “Luc8” while this protein was referred in section 3.2, line 217.

3)      Incorrect information in Line 139, “It is the smallest luciferase to be characterized at 19 kDa …..” However, TurboLuc luciferase was claimed to be the smallest luciferase reporter enzyme described to date (16 kDa) by Auld DS., et al. (Biochemistry 2018, 57(31), 4700).

Table 1 is not informative but rather causing confusion. For example,

1)      Line 79, “….oxygen and NADPH as cofactor,…”; Those cofactors were not presented in Table 1, under “Co factor” column.

2)      Line 89, “….requires the presence of co-factors, oxygen and magnesium…”. Those cofactors were not presented in Table 1, under “Co factor” column.

As authors quoted “quantum yield” to determine the efficiency of luciferase multiple times throughout the manuscript, this characteristic would be useful to be included in Table 1.

In Table1, what is the exact definition of “Type” (the first column heading)?

The names of luciferases in Table 1 are not scientific. It should be presented as exact name of Species.

Line 88, “562 kDa” should be “562 nm”

First paragraph under section 3.1, through lines 148-153, is poorly described and not relevant to the subject category, “QD-based BRET sensor for detecting biomolecules”. This research is not detecting spermatozoa rather imaging it to track the movement, viability, and fertilizing potential of labeled spermatozoa. This research is also claimed as the first reported BRET-QO labeled- spermatozoa but appropriate emphasis was not granted.

Author Response

Authors would like to express the great appreciation to the editor and all the reviewers for their time and constructive comments on our manuscript (biosensors-449769), entitled “Integration of Nanomaterials and Bioluminescence Resonance Energy Transfer Techniques for Sensing Biomolecules”. We have revised the manuscript carefully per all the review comment. Detailed response can be found in the following point-to-point responses, where different colors are used to distinguish the comments (in black) from the corresponding responses (in blue).

The manuscript entitled “Integration of Nanomaterials and Bioluminescence Resonance Energy Transfer Techniques for Sensing Biomolecules” by Ji Su Song and Jin Zhang presented BRET-QD technology and its application in biomaterial sensing and imaging. In detail, authors described some of bioluminescent proteins as a source of light donor. Applications such as detection of DNA mismatch, small molecule and apoptotic cells were illustrated as well as in vivo and in vitro imaging of spermatozoa, tumors, lymphatic. However, this manuscript shows lack of accuracy and the information is not comprehensive to meet the standard for publication to Biosensors as a “Review”. 

Figures 1 and 2 and the corresponding description are very similar with previous review presented by Fei Ma, et al. (Figures 7 and 8; J. Mater. Chem. B, 2018, 6, 6173-6190).

Authors’ response: The authors acknowledge that there is an overlap in the scope of both review papers. We obtained the copyright permit on using Figures. Also, we revised the manuscript to address the bioluminescence technique,

This manuscript poorly referenced in context of bioluminescent proteins, in particular Table 1 such as

1)       The list of luciferases is not comprehensive compared with a list presented by Gokhin S., et. al. in 2017 (http://dx.doi.org/10.1039/9781782626770-00117).

Authors’ response: We are thankful for the reviewer’s comment. Unfortunately, the above paper was not accessible from our institution; however, more appropriate review papers were included in the revised version for comparison.

2)       Author failed to include a luciferase, “Luc8” while this protein was referred in section 3.2, line 217.

Authors’ response: Luc8 is an eight-mutation variant of Rluc. This information was added to the paper accordingly.

3)       Incorrect information in Line 139, “It is the smallest luciferase to be characterized at 19 kDa …..” However, TurboLuc luciferase was claimed to be the smallest luciferase reporter enzyme described to date (16 kDa) by Auld DS., et al. (Biochemistry 2018, 57(31), 4700).

Authors’ response: The wording was fixed to “one of the smallest luciferase” and “one of the brightest bioluminescence”. TurboLuc was not included in the list due to lack of studies (only 2 results with “Turboluc” can be found on PubMed).

Table 1 is not informative but rather causing confusion. For example,

1)       Line 79, “….oxygen and NADPH as cofactor,…”; Those cofactors were not presented in Table 1, under “Co factor” column.

Authors’ response: The “Co factor” column was removed from the table as we decided that reading the content would clarify the use of cofactors for each protein.

2)       Line 89, “….requires the presence of co-factors, oxygen and magnesium…”. Those cofactors were not presented in Table 1, under “Co factor” column.

Authors’ response: It was revised. See the above response.

As authors quoted “quantum yield” to determine the efficiency of luciferase multiple times throughout the manuscript, this characteristic would be useful to be included in Table 1.

Authors’ response: Quantum yield was not included in the table as its quantification is rather relative than absolute. It would be better for readers to read the content on the information of the quantum yield.

In Table1, what is the exact definition of “Type” (the first column heading)?

Authors’ response:  Table 1 is revised. “Type” column was removed to make the information clearer.

The names of luciferases in Table 1 are not scientific. It should be presented as exact name of Species.

Authors’ response: It was corrected as per requested.

Line 88, “562 kDa” should be “562 nm”

Authors’ response: It was corrected as per requested.

First paragraph under section 3.1, through lines 148-153, is poorly described and not relevant to the subject category, “QD-based BRET sensor for detecting biomolecules”. This research is not detecting spermatozoa rather imaging it to track the movement, viability, and fertilizing potential of labeled spermatozoa. This research is also claimed as the first reported BRET-QO labeled- spermatozoa but appropriate emphasis was not granted.

Authors’ response: The authors acknowledge the irrelevancy of the research to the subject category and have removed it from the section.

Reviewer 2 Report

Strong Points:

- The properties and advantages of BRET which could have advantages compared with FRET.

Weak Points:

- The properties of the matching QDs with the bioluminescent donors should be described in detail, for example in terms of molecular size, chemical bonding or conjugating process to the donor, the emission strength of the QDs, the required number of QDs to emit a visible light.

- The comparison with classical FRET should be done in terms of the required number of bioluminescent donor molecules and QDs, such as in a table. For example, in the classical FRET, powerful  laser can excite a specific amount of donors to realize flourescence. How much  molecules do we need here? Is not there any disadvantage for such a process? Another table also for the BRET duration and strength could be included.

- Section 4 could be renamed as "Biocompatibility" or a new section could be included with this name.

- Theoretical modeling of BRET and  Fluorescence by unbound excitation from luminescence (FUEL) could be included and compared with sixth order distance dependent FRET modeling to better explain the reader the theoretical basics of these three mechanisms.

- In the introduction in a small paragraph or as a small new subsection can be included where FRET can be replaced with BRET for sensor applications. It will provide the reader more potential applications of BRET: such as "Theoretical Modeling of Viscosity Monitoring with Vibrating Resonance Energy Transfer" in MDPI Micromachines, "Fluorescence resonance energy transfer (FRET)-based biosensors: visualizing cellular dynamics and bioenergetics" in Springer etc.

- Some writing errors such as: "NIR light cannot penetrated"  or  "CdSe/ZS“

- Figure 6 pictures are not clearly visible with some blur.

Author Response

Authors would like to express the great appreciation to the editor and all the reviewers for their time and constructive comments on our manuscript (biosensors-449769), entitled “Integration of Nanomaterials and Bioluminescence Resonance Energy Transfer Techniques for Sensing Biomolecules”. We have revised the manuscript carefully per all the review comment. Detailed response can be found in the following point-to-point responses, where different colors are used to distinguish the comments (in black) from the corresponding responses (in blue).

Strong Points:

- The properties and advantages of BRET which could have advantages compared with FRET.

Authors’ response: We appreciate the reviewer for the positive comment.

Weak Points:

- The properties of the matching QDs with the bioluminescent donors should be described in detail, for example in terms of molecular size, chemical bonding or conjugating process to the donor, the emission strength of the QDs, the required number of QDs to emit a visible light.

Authors’ response: We have included information pertaining to how QDs are used and functionalized in BRET sensors

- The comparison with classical FRET should be done in terms of the required number of bioluminescent donor molecules and QDs, such as in a table. For example, in the classical FRET, powerful laser can excite a specific number of donors to realize fluorescence. How much molecules do we need here? Is not there any disadvantage for such a process? Another table also for the BRET duration and strength could be included.

Authors’ response: We appreciate the reviewer’s technique insight. The comparation of FRET and BRET can be found in Page 2. This review focuses on the BRET sensor by applying quantum dots (QDs). We introduce the major bioluminescence protein here following the progress of QD-based BRET sensor. We hope this review will help audience quickly control the current development of this new type of biosensor.

- Section 4 could be renamed as "Biocompatibility" or a new section could be included with this name.

Authors’ response: We are thankful for the reviewer’s comment. We have included a new section specifically for biocompatibility.

- Theoretical modeling of BRET and  Fluorescence by unbound excitation from luminescence (FUEL) could be included and compared with sixth order distance dependent FRET modeling to better explain the reader the theoretical basics of these three mechanisms.

Authors’ response: The authors appreciate the suggestion, however upon further deliberation, we have decided to remove the FUEL section as it does not have significant relevance to the review paper’s scope in terms of quantum dots-based BRET sensors. We have taken into consideration the inclusion of the theoretical basis for FRET and BRET and included it in the introduction

- In the introduction in a small paragraph or as a small new subsection can be included where FRET can be replaced with BRET for sensor applications. It will provide the reader more potential applications of BRET: such as "Theoretical Modeling of Viscosity Monitoring with Vibrating Resonance Energy Transfer" in MDPI Micromachines, "Fluorescence resonance energy transfer (FRET)-based biosensors: visualizing cellular dynamics and bioenergetics" in Springer etc.

Authors’ response: The authors appreciate the suggestion and have added it to the paper

- Some writing errors such as: "NIR light cannot penetrated” or” CdSe/ZS “

Authors’ response: The authors acknowledge and have corrected the errors.

- Figure 6 pictures are not clearly visible with some blur.

Authors’ response: We have included an image of the figure with higher resolution and replaced the original image.

Round  2

Reviewer 1 Report

The authors have tried hard to address reviewers’ comments and supplied additional information.  I recommend publishing this manuscript after a few of minor correction.

A paragraph, lines 74-85, is repeatedly explaining the advantage of BRET over FRET (or luminescence over fluorescence) which was described at the beginning (lines 29-41). Also the reference number is not right on line 79 and maybe line 82.

Figure numbers need to be revised throughout the manuscript.

Author Response

We appreciate the comments of reviewer. The point-by-point response is as follows;

A paragraph, lines 74-85, is repeatedly explaining the advantage of BRET over FRET (or luminescence over fluorescence) which was described at the beginning (lines 29-41). Also the reference number is not right on line 79 and maybe line 82.

Response. The sentences were revised to make it concisely. The reference No. is checked.

Figure numbers need to be revised throughout the manuscript.

Response. We have checked and corrected the figure numbers.